# Circulating Exosomal-DNA in Glioma Patients: A Quantitative Study and Histopathological Correlations—A Preliminary Study

**DOI:** 10.3390/brainsci12040500

**Published:** 2022-04-14

**Authors:** Amedeo Piazza, Paolo Rosa, Luca Ricciardi, Antonella Mangraviti, Luca Pacini, Antonella Calogero, Antonino Raco, Massimo Miscusi

**Affiliations:** 1Operative Unit of Neurosurgery, Department of NESMOS, Sapienza University of Rome, 00185 Rome, Italy; luca.ricciardi@uniroma1.it (L.R.); antomangraviti@gmail.com (A.M.); antonino.raco@uniroma1.it (A.R.); massimo.miscusi@uniroma1.it (M.M.); 2Department of Medical-Surgical Sciences and Biotechnologies, Sapienza University of Rome, 04100 Latina, Italy; p.rosa@uniroma1.it (P.R.); antonella.calogero@uniroma1.it (A.C.); 3Pathology Unit, I.C.O.T. Hospital, 04100 Latina, Italy; pacini.luca@gmail.com

**Keywords:** glioma, exosome, liquid biopsy, EVs

## Abstract

Glial neoplasms are a group of diseases with poor prognoses. Not all risk factors are known, and no screening tests are available. Only histology provides certain diagnosis. As already reported, DNA transported by exosomes can be an excellent source of information shared by cells locally or systemically. These vesicles seem to be one of the main mechanisms of tumor remote intercellular signaling used to induce immune deregulation, apoptosis, and both phenotypic and genotypic modifications. In this study, we evaluated the exosomal DNA (exoDNA) concentration in blood samples of patients affected by cerebral glioma and correlated it with histological and radiological characteristics of tumors. From 14 patients with diagnosed primary or recurrent glioma, we obtained MRI imaging data, histological data, and preoperative blood samples that were used to extract circulating exosomal DNA, which we then quantified. Our results demonstrate a relationship between the amount of circulating exosomal DNA and tumor volume, and mitotic activity. In particular, a high concentration of exoDNA was noted in low-grade gliomas. Our results suggest a possible role of exoDNAs in the diagnosis of brain glioma. They could be particularly useful in detecting early recurrent high-grade gliomas and asymptomatic low-grade gliomas.

## 1. Introduction

Glial neoplasms are a group of diseases with poor prognoses. Not all risk factors are known, and no screening tests are available. In most cases, diagnosis is late, and due to the appearance of neurological signs and symptoms. Only the histology of surgical specimens can confirm the diagnosis and characterize the molecular pattern of glioma cells.

DNA transported by exosomic vesicles can be an excellent source of information because protected by the vesicles, it does not degrade. These vesicles seem to represent one of the main mechanisms of tumor remote intercellular signaling used to induce immune deregulation, apoptosis, and both phenotypic and genotypic modifications [1,2].

In Glioblastoma multiforme (GBM) patients, the concentration of plasma extracellular vesicles (EVs) is increased in comparison to healthy controls and patients with other CNS diseases, and the EV concentration over different time points is correlated with tumor recurrence, suggesting that exosomes could help to predict GBM recurrence [3]. 

In this study, we quantified exosomal DNA (exoDNA) in plasma from glioma and glioblastoma patients and investigated its possible correlation with histological and radiological tumor characteristics.

## 2. Materials and Methods

### 2.1. Study Design

This is a prospective observational study. The institutional review board approved the investigation and the anonymous collection of patients’ data for scientific purposes. 

### 2.2. Patient Population

A series of 14 patients operated for intracranial glioma at Sant’Andrea University Hospital of Rome within the period from January 2019 to May 2021 were prospectively enrolled in our study. Before surgery, all patients were studied with brain MRI (Magnetic Resonance Imaging), including T1 sequences pre and post gadolinium enhancement and T2-FLAIR sequence, and a 10-mL blood sample was collected for molecular biology experiments. Histological analysis was conducted to evaluate the Ki-67 and mitotic index quantitatively.

The patients were categorized into three subgroups according to clinical and histological parameters (according to WHO 2016 classification): (1) newly diagnosed glioblastoma, (2) recurrent glioblastoma, and (3) low-grade glioma. Patients in subgroup 2 had a previous history of surgery for tumor removal and follow-up CT and RT according to the STUPP protocol.

A series of 10 patients scheduled for brain MRI and blood samples for non-specific cephalea at our neurologic department constituted the control group. Their images were retrieved anonymously from the institutional PACS viewer, and their stored blood samples were processed for molecular biology experiments. No modifications to their study protocol were necessary for conducting these evaluations. Subjects included in the control group were MRI-negative for intracranial masses and had no previous history of systemic neoplasms.

### 2.3. Magnetic Resonance Imaging

Radiological studies were conducted on all patients according to the glioma-and-neurodegenerative disorders study protocol, as standardized at the Neuroradiology Department, using the same MRI sequences for all patients. All images were evaluated by a single senior neuroradiologist of the same institution.

The analysis of T2-weighted FLAIR and post-gadolinium T1 sequences allowed the measurement of tumor volume (cm^3^) in, respectively, low-grade and high-grade gliomas. In high-grade tumors, hypointense volumes (cm^3^), corresponding to necrotic tissue, were also evaluated (Figure 1 and Figure 2).

### 2.4. Blood Sample 

Before surgery, blood samples were collected by venous sampling from the cubital fossa. The plasma was extracted at a controlled temperature (4 °C) by serial centrifugation.

An exoEasy Maxi kit (Qiagen, Germantown, MD, USA) was used to isolate exosomes from the plasma. A QIAamp MinElute Virus Spin Kit (Qiagen, Germantown, MD, USA) was used to extract the DNA contained in the exosomes (exoDNA). A fluorometer (Qubit 4, Invitrogen, Waltham, MA, USA) was used to measure the exoDNA concentration (in ng/µL), following the manufacturer’s instructions. 

### 2.5. Statistical Analysis 

The values were reported as mean ± standard deviation (SD). Student’s *t*-test was used to compare the continuous quantitative variables. Fisher’s exact test (2-sided) was used to compare the categorical variables. Statistical significance was predetermined at an alpha value of 0.05. Univariate and multivariate multiple regression analyses were performed. AnalystSoft Inc. StatPlus 2020 (Walnut, CA, USA) was used for data analysis.

## 3. Results

In the study group, the mean age was 43 (±21.3), and the male-to-female ratio was 3:2. There were 10 patients who had histological diagnoses of glioblastoma WHO grade IV, 1 with glioma WHO grade III, 3 with R132H IDH1-mutated WHO grade II diffuse astrocytoma, and 2 with a diffuse astrocytoma that was IDH1-negative WHO grade II. We then analyzed, by immunohistochemistry, the positivity of the marker Ki-67 and calculated the mitotic index to evaluate cell proliferation within the tumor mass. In subgroup 1, the mean expression of Ki-67 was 40 ± 20%, and the mean expression of the mitotic index was 31 ± 18.31/10 HPF. In subgroup 2, the mean expression of Ki-67 was 40 ± 30%, and the mean expression of the mitotic index was 19.33 ± 11.01/10 HPF. In subgroup 3, the mean expression of Ki-67 was 3 ± 2.7%, and the mean expression of the mitotic index was 3.5 ± 2.1/10 HPF (Table 1).

We also analyzed the mean hyperintense and hypointense volumes in order to have information about active or necrotic tumor areas. In subgroup 1, the mean hyperintense volume was 5.66 ± 2.9 cm^3^, the mean hypointense volume was 18.2 ± 8.9 cm^3^, and the mean total tumor volume was 23.88 ± 12.9 cm^3^. In subgroup 2, the mean hyperintense volume was 9.42 ± 4.1 cm^3^, the mean hypointense volume was 1.1 ± 0.7 cm^3^, and the mean total tumor volume was 10.53 ± 3.6 cm^3^. In subgroup 3, the mean tumor volume, calculated from the T2-FLAIR sequences, was 18.06 ± 4.4 cm^3^ (Table 1).

Finally, we analyzed the quantity of DNA extracted from plasma-isolated exosomes. The mean plasma exosome concentrations were 10.38 ± 5.54 ng/µL in subgroup 1, 8.55 ± 4.8 ng/µL in subgroup 2, and 82.1 ± 13.6 ng/µL in subgroup 3. In the control group, the mean concentration of exosomes was 11.25 ± 4.2 ng/µL. 

The mean amounts of exoDNA per unit of tumor volume were 0.433 ng/µL/cm^3^ in subgroup 1, 0.811 ng/µL/cm^3^ in subgroup 2, and 5.831 ng/µL/cm^3^ in subgroup 3 (Table 1).

### Data Analysis

In subgroup 1, exoDNA and total tumor volume (correlation index = −0.344, *p*-value = 0.0118) were inversely correlated with the hypointense volumes in RM gh-T1 sequences (correlation index = −0.474, *p*-value = 0.0426). No correlation between exoDNA concentration and mitotic index was found.

In subgroup 2, a linear correlation was observed between the exoDNA content and both the total tumor volume (correlation index = 0.473, *p*-value = 0.5267) and the hyperintense volume in gh-T1 (correlation index = 0.347, *p*-value = 0.5267). There was a strong correlation (correlation index = 0.742, *p*-value = 0.2579) between the amount of DNA and the mitotic index.

Analyzing groups 1 and 2, a linear correlation was observed between Ki-67 expression and the percentage of hyperintense volume on gh-T1 sequences with respect to the total tumor volume (correlation index = 0.733, *p*-value = 0.0103). 

Group 3 showed high levels of exoDNA in both the blood (82.1 ± 13.6 ng/µL) and the tumors (5.831 ng/µL/cm^3^) (Table 1).

## 4. Discussion

Liquid biopsy is a blood sampling method used to detect circulating cancer cells or their genetic material. It is playing an increasingly important role in the diagnostic and prognostic evaluation of neoplasms. Numerous studies have shown that the amount of free circulating DNA (cirDNAfree) in the blood increases in relation to the total number of circulating cancer cells [4] and the tumor volume measured by CT [5]. High levels of cirDNAfree have been found in the blood of patients with advanced and metastatic carcinoma compared to that of patients with early-stage cancer [6,7]. For example, in gastric cancer, it was observed that cirDNAfree values were increased in patients with cancer (5–1500 ng/mL) compared with the levels in healthy individuals (1–5 ng/mL) [8,9]. The study by Li et al. [10] clearly demonstrated that the release of cirDNAfree is closely related to cell death, as demonstrated by in vitro experiments in which the peak of cirDNAfree found in the supernatant coincided with the peak of necrosis. Interestingly, cirDNAfree in the blood and the CSF of patients with glioma has been correlated with tumor aggressiveness, suggesting its use as a prognostic marker in gliomas [11].

Nevertheless, circulating DNA in blood can also be found in microvesicles and in exosomes. Exosomes are extracellular vesicles with a diameter of around 30–100 nm and a bilayer membrane [12,13]. Exosome cargo comprises proteins (including enzymes and transcription factors), DNA fragments, mRNAs, micro-RNAs, and long non-coding RNAs (lncRNAs) [14,15]. Exosomes are critical for cellular signaling in normal physiology and in pathological conditions, most notably in cancer. They participate in the modification of the tumor microenvironment, angiogenesis, metastasis, and immunity evasion [16]. To date, various notable studies have been conducted to assess the utility of exosomes as diagnostic and prognostic biomarkers, but they have been restricted to exosomal microRNAs (miRs) [17,18]. Exosomal miR-18, miR-301, and miR-124 are potential diagnostic biomarkers for human glioma [19,20,21].

Exosomes also contain DNA molecules representing introns, non-coding genomic DNA, and other chromosomal sequences, including transposable elements. Several mutated genes specific to brain glioma have been found in exosomes, such as MGMT [22,23,24], PTEN [22], EGFR [25,26], and many others [27]. We aimed to verify whether the exoDNA concentration in the plasma of patients affected by glioma was related to the volume and histological grading of their tumors.

Our data suggest that in patients with low-grade glioma, where necrotic areas are not present and the mitotic index is low, and also in patients with small tumor volumes, the amount of circulating exoDNA is directly correlated to the total tumor volume. The inverse correlation between exoDNA and cell death has already been demonstrated for other tumors [28,29]. In this context, the absence of a necrotic area could suggest that all glioma cells are involved in exoDNA signaling; therefore, in the first phase of tumor growth, exoDNA release could be a means by which low-grade tumor cells create local and systemic conditions that facilitate progression. For instance, it has been demonstrated that glioma cells might stimulate angiogenesis by transferring linc-CCAT2 and linc-POU3F3 to endothelial cells through exosomes [30,31]. Janusz rak et al. [32] demonstrated that glioma cells could transfer the EGFRvIII receptor to other cells through exoDNA, inducing pro-oncogenic modifications of the MAPK and AKT signaling pathways. Finally, exosomes released by glioma cells could have regulatory effects on the immune system, both locally and at a distance: for example, they seem to be able to modify B-cell activity through PIGFs [2,4].

Conversely, in high-grade glioma, the exoDNA concentration in plasma is inversely correlated to tumor volume and to the mitotic index and Ki-67. The extension of the necrotic area and the consequent reduction in exosome-releasing cells could explain the inverse relation with the total tumor volume, while the reduced release of exoDNA from highly mitotic cells could be due to the end of the tumor’s need to escape immune surveillance once it has acquired an evidently malignant phenotype.

In patients with high-grade glioma recurrence (subgroup 2), we observed a linear correlation between total tumor volume and exoDNA concentration, similar to what we observed in patients with low-grade glioma, although these two groups present clear differences in mitotic index and Ki-67 positivity. This could demonstrate that after surgery, chemotherapy, and radiotherapy, residual glioma cell or newly formed glioma cells behave as low-grade cells. They could prepare a local and systemic milieu for regrowth by using exoDNA signaling after a sort of biological reset obtained by surgical and medical therapy. This study is, in no small part, limited by the lack of qualitative assessment of exoDNA, as well as by the small patient sample; however, it is one of the first preliminary analyses of exoDNA across different histopathological and clinical glioma subtypes. Considering the well-known genetic heterogeneity underlying distinctions between primary vs. secondary and low vs. high-grade gliomas, an exoDNA analysis of such wide breadth as presented here is likely to be setting the stage for any future study looking to identify specific mutations. This development would markedly widen the diagnostic and prognostic role of liquid biopsies in patients affected by glioma. 

This preliminary study clearly anticipates our intent to analyze exoDNA mutations by Next-Generation Sequencing with the purpose of finding any correlations with those of the tumoral tissue. In light of these considerations, the methodology we explored could be a valuable tool for all the subgroups analyzed with the unique aim to diagnose, predict, and/or monitor tumors by non-invasive procedures.

## 5. Conclusions

Our study seems to confirm the important role of exosomes in the oncogenesis and progression of human glioma. The concentration of exoDNA in the plasma of our patients varied across the different stages of tumor growth and is, therefore, a potentially useful marker in the early diagnosis of new cases or during the disease-free interval in patients who have undergone primary treatment. Further studies involving a larger series of patients are expected to verify the role that exosomes may have in the pathogenesis and progression of brain gliomas. 

## Figures and Tables

**Figure 1 brainsci-12-00500-f001:**
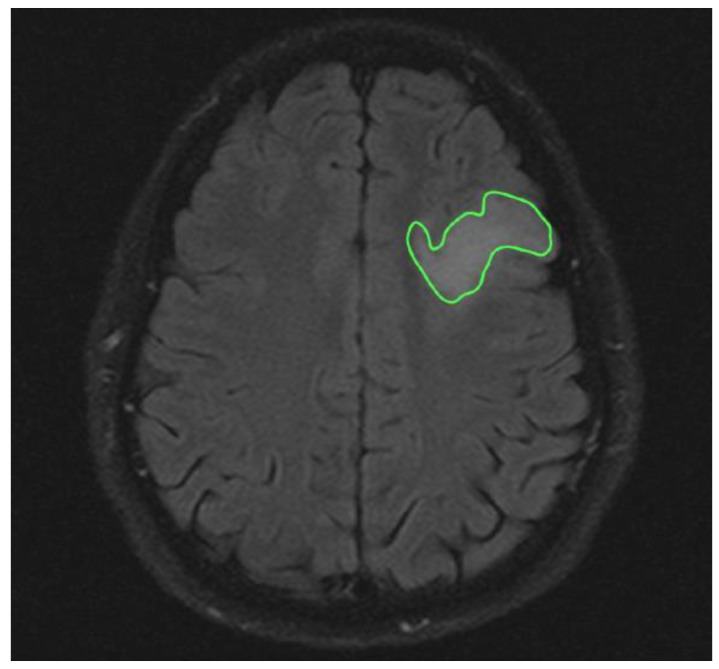
Axial ROI of low-grade glioma in T2 Flair sequences.

**Figure 2 brainsci-12-00500-f002:**
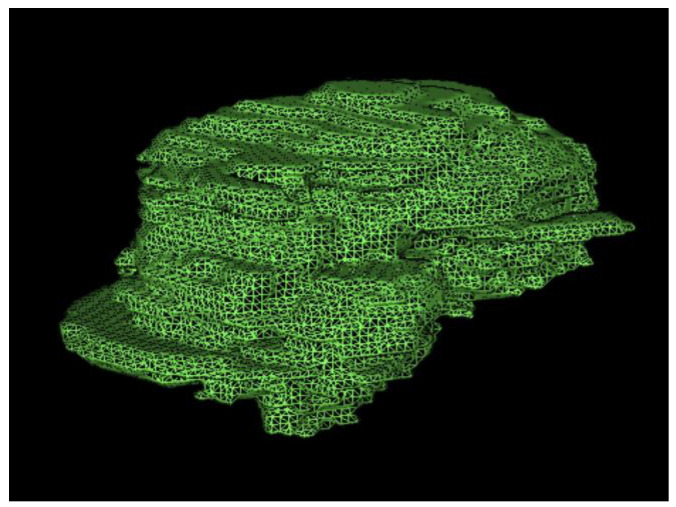
Three-dimensional rendering of total tumor volume in a patient treated for HGG.

**Table 1 brainsci-12-00500-t001:** Summary of Patients Characteristics.

Characteristics	N°
Patients	16
Age	43 ± 21.3
Male female ratio	3:2
Groups	n°
Group 1	7
Group 2	4
Group 3	5
Group control	10
Ki—67andMitoticindex	mean values
Group1 Ki-67	40 ± 20 percent
Mitotic Index	31 ± 18.31/10 HPF
Group2 Ki-67	40 ± 30 percent
mitotic index	19.33 ± 11.01/10 HPF
Group3 Ki-67	3 ± 2.7 percent
Mitotic index	3.5 ± 2.1/10 HPF
MRiTumorvolumes	mean values
Group 1 MRi gh-t1 hyperintense volume	5.66 ± 2.9 cm^3^
MRi gh-t1 hypointense volume	18.2 ± 8.9 cm^3^
MRi total tumor volume	23.88 ± 12.9 cm^3^
Group 2 MRi gh-t1 hyperintense volume	9.42 ± 4.1 cm^3^
MRi gh-t1 hypointense volume	1.1 ± 0.7 cm^3^
MRi total tumor volume	23.88 ± 12.9 cm^3^
Group 3 Tumor volume MRi Flair	18.06 ± 4.4 cm^3^
Exosomeconcetration	mean values
Group 1	10.38 ± 5.54 ng/µL
Group 2	8.55 ± 4.8 ng/µL
Group 3	82.1 ± 13.6 ng/µL
Group Control	11.25 ± 4.2 ng/µL
DNAexo—totaltumorvolumeRatio	mean values
Group 1	0.433 ng/µL/cm^3^
Group 2	0.811 ng/µL/cm^3^
Group 3	5.831 ng/µL/cm^3^

## Data Availability

Not applicable.

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
