# Peer review of "Circulating Exosomal-DNA in Glioma Patients: A Quantitative Study and Histopathological Correlations—A Preliminary Study"

_brainsci, 2022, doi:10.3390/brainsci12040500_

Round 1

Reviewer 1 Report

The manuscript proposed by Piazza et al. Is a preliminary study on circulating exosomal-DNA in glioma patients. They performed quantitation  of exosomes and correlated this information to histopathological data. The experiments were well performed but data could be better described and the manuscript must be improved before to be published:

  • As mentioned by the authors in the Conclusion, only a small series of patients was studied. The panel should be enlarged in order to conclude anything; Especially with patients from the Group 3 where they were only 3 patients. In this group, concentration of exosome from plasma and the amount of exoDNA per unit of tumor volume are particularly high and this has to be confirmed before to propose such an evaluation as potential marker for diagnosis.
  • Concerning the proposal to use such a value as diagnostic/prognostic market, the author should explain more the way they propose to use practically this. In this study, they used mass spectrometry for measuring exoDNA. Is this method feasible in all hospitals ? how to standardize the method ? I am not sure it is realistic.
  • All the end of the Discussion for tenting to rely exosome and cell death is hypothetical. The authors based only their discussion on the few data they presented but they never refer to any other works that could support their explanation.
  • Overall, the literature cited is far from exhaustive about exosome and glioma. For example, the review of Ghaemmaghami et al. (Cell Commun Signal. 2020) and the review from Peng et al. (Front. Immunol. 2022) could be mentioned. They both detailed the major role of exosomal miRNA in glioma. A point largely explored but never mentioned in the Introduction and, at least, as important as exoDNA. In terms of exosomal DNA, another review from Sharma & Johnson (J. Cell. Physiol., 2020), dedicated specifically to exosome DNA in terms of diagnostic marker could be considered (and its many references). Finally, the review of Vaidya & Kiminobu (Genes, 2020) especially on this marker in glioma should be mentioned.
  • The advantage to useDNA as biomarker, is the possibility to sequence it. As developed in the review above, it is now difficult to consider only the amount of exoDNA without the benefit to provide highly specific information on tumor cells by the sequencing.

Author Response

The manuscript proposed by Piazza et al. Is a preliminary study on circulating exosomal-DNA in glioma patients. They performed quantitation of exosomes and correlated this information to histopathological data. The experiments were well performed but data could be better described and the manuscript must be improved before to be published:

  • As mentioned by the authors in the Conclusion, only a small series of patients was studied. The panel should be enlarged in order to conclude anything; Especially with patients from the Group 3 where they were only 3 patients. In this group, concentration of exosome from plasma and the amount of exoDNA per unit of tumor volume are particularly high and this has to be confirmed before to propose such an evaluation as potential marker for diagnosis.
  • We thank the reviewer for his suggestions. We promptly analyzed the exoDNA from other two patients of group 3 that were added in the main text of the article. We are aware that the series of patients is still small, in fact we highlighted this point in the conclusion as a limit of the study. However, we discussed our observations hypothesizing that high amount of exoDNA are found in patients with low-grade glioma as a consequence of low levels of cell death and added references.

  • Concerning the proposal to use such a value as diagnostic/prognostic market, the author should explain more the way they propose to use practically this. In this study, they used mass spectrometry for measuring exoDNA. Is this method feasible in all hospitals? how to standardize the method? I am not sure it is realistic.
  • We kindly thank the referee for raising this question. Unfortunately, there was an error in material and methods part and we feel terribly sorry. We wrongly reported that we recurred to a mass spectrophotometer to quantify exoDNA. This is not correct, and we promptly corrected this section with this sentence: ‘’A fluorometer (Qubit 4, Invitrogen) was used to measure the exoDNA concentration (in ng/ul), by following the manufacturer’s instructions’’.

  • All the end of the Discussion for tenting to rely exosome and cell death is hypothetical. The authors based only their discussion on the few data they presented but they never refer to any other works that could support their explanation.
  • We are terribly sorry to have not reported any references supporting our observations and hypotheses. We included in the text the references (Takahashi et al., Nature Communications, 2017) and (Sharma et al., Scientific Reports, 2020). Although this has been demonstrated for other tumors, our preliminary observations could be plausible to exist.

  • Overall, the literature cited is far from exhaustive about exosome and glioma. For example, the review of Ghaemmaghami et al. (Cell Commun Signal. 2020) and the review from Peng et al. (Front. Immunol. 2022) could be mentioned. They both detailed the major role of exosomal miRNA in glioma. A point largely explored but never mentioned in the Introduction and, at least, as important as exoDNA. In terms of exosomal DNA, another review from Sharma & Johnson (J. Cell. Physiol., 2020), dedicated specifically to exosome DNA in terms of diagnostic marker could be considered (and its many references). Finally, the review of Vaidya & Kiminobu (Genes, 2020) especially on this marker in glioma should be mentioned.
  • We appreciated the suggestions of the referee and we added all the references in the main text.

  • The advantage to useDNA as biomarker, is the possibility to sequence it. As developed in the review above, it is now difficult to consider only the amount of exoDNA without the benefit to provide highly specific information on tumor cells by the sequencing.
  • We totally agree with the referee that we thank for this precious observation. In fact, our preliminary quantitative analysis was performed and reported with the solely aim to understand if such a qualitative analysis on DNA sequencing could be affordable. In the light of the results of this study, we have already planned to recur to NGS in order to analyze a panel of genes related to glioma with the greatest sensibility due to the low amount of DNA found in patients of groups 1-2. At the moment, this goes beyond the aim of the present study, which remains preliminary and explorative.

Reviewer 2 Report

I would like to see the following points improved and clarified:

  1. Explain the small number of patients
  2. Why were MR metabolic sequences such as perfusion and spectroscopy not used?
  3. Better explain the results
  4. Improve the discussion, namely with a more exhaustive comparison with the results of the already published literature
  5. Discuss separately the importance of liquid biopsies in the 3 groups studied
  6.  

Author Response

I would like to see the following points improved and clarified:

  1. Explain the small number of patients

  • We thank the referee for the suggestions. As already mentioned, the small number of patients come from the nature of the present study, which is preliminary, and speculative on a methodology that could be seen as expensive but can give the advantage to predict and/or monitor tumors as gliomas without recurring to invasive procedures.

2. Why were MR metabolic sequences such as perfusion and spectroscopy not used?

  • We understand the question raised by the referee. Unfortunately, we could only focus on the volume measured by T2-weighted FLAIR and post-gadolinium T1 sequences.

3. Better explain the results

  • We are sorry to have not explained exhaustively the results. Now we explained better those results that could be hard to understand for the reader.

4. Improve the discussion, namely with a more exhaustive comparison with the results of the already published literature

  • We totally agree with the observations raised from the referee and we promptly added more references supporting our observations.

5. Discuss separately the importance of liquid biopsies in the 3 groups studied

  • We thank the referee for the suggestions. We discussed better the importance of liquid biopsies in the groups studied by adding more references supporting our observations and by highlighting the urgent need to recur to non-invasive procedures for brain tumors.

Reviewer 3 Report

Dear Authors,

This quantitative study is well planned and the manuscript is comprehensive. Please include some histology images to justify the title itself. 

Good Luck!.

Author Response

Dear Authors,

This quantitative study is well planned and the manuscript is comprehensive. Please include some histology images to justify the title itself.

Good Luck!

We thank the referee for the encouragement. Unfortunally, we cannot add representative histology images at the moment. But we are sure that the title is still supported anyways by the analysis we performed on histological sections (e.g. Ki-67).

Round 2

Reviewer 1 Report

With this version, the authors significantly ameliorated the manuscript by better explaining the Methods, adding new patients and extending the Discussion. Nevertheless, the last sentence of the Conclusion should be corrected. Moreover,  as reply to a referee, the future sequencing by NGS in order to provide more information from such a study should be specified in the Discussion, in order to better place this preliminary study in a global investigation with a larger expected goal.

Author Response

We thank the referee for his/her comments. We better explained in the last part of the discussion the preliminary nature of this study anticipating to the reader our intent to analyze by NGS glioma mutations in exosomes.

In the conclusions we promptly deleted the last part which is as an error from the last review process.